# Advanced RuO_2_ Thin Films for pH Sensing Application

**DOI:** 10.3390/s20226432

**Published:** 2020-11-11

**Authors:** Xinyue Yao, Mikko Vepsäläinen, Fabio Isa, Phil Martin, Paul Munroe, Avi Bendavid

**Affiliations:** 1School of Materials Science and Engineering, University of New South Wales, Kensington, NSW 2052, Australia; Xinyue.yao1@unswalumni.com (X.Y.); p.munroe@unsw.edu.au (P.M.); 2CSIRO Mineral Resources, P.O. Box 312, Clayton South, VIC 3169, Australia; mikko.vepsalainen@csiro.au; 3CSIRO Manufacturing, P.O. Box 218, 36 Bradfield Road, Lindfield, NSW 2070, Australia; fabio_Isa@outlook.com (F.I.); phil.martin@csiro.au (P.M.)

**Keywords:** ruthenium dioxide, magnetron sputtering conditions, thin film characterisation, pH performance

## Abstract

RuO_2_ thin films were prepared using magnetron sputtering under different deposition conditions, including direct current (DC) and radio frequency (RF) discharges, metallic/oxide cathodes, different substrate temperatures, pressures, and deposition times. The surface morphology, residual stress, composition, crystal structure, mechanical properties, and pH performances of these RuO_2_ thin films were investigated. The RuO_2_ thin films RF sputtered from a metallic cathode at 250 °C exhibited good pH sensitivity of 56.35 mV/pH. However, these films were rougher, less dense, and relatively softer. However, the DC sputtered RuO_2_ thin film prepared from an oxide cathode at 250 °C exhibited a pH sensitivity of 57.37 mV/pH with a smoother surface, denser microstructure and higher hardness. The thin film RF sputtered from the metallic cathode exhibited better pH response than those RF sputtered from the oxide cathode due to the higher percentage of the RuO_3_ phase present in this film.

## 1. Introduction

The sensing of pH is very important in several chemical and biological processes, such as for applications in water and food quality monitoring, and wearable systems for chronic diseases [1]. In order to optimise the desired response and to eliminate unwanted reactions, pH measurement, and control are both required in many applications, such as blood monitoring, environmental monitoring, water quality monitoring, and various clinical tests. The glass electrode sensor has been commonly used for pH measurement due to its high accuracy, fast response, and ideal Nernst behaviour. However, with the increasing requirements for different applications, glass electrodes exhibit drawbacks such as instability in high temperature environments, poor mechanical properties in high pressure environments, and difficulties in miniaturisation [1]. Therefore, many other pH measurement techniques have been developed recently, one of which is the solid-state metal oxide thin film sensor, based on oxides such as PtO_2_, IrO_2_, RuO_2_, SnO_2_, and Ta_2_O_5_, that has been demonstrated to exhibit excellent pH sensing performance at high temperatures and pressures and these are promising candidates for a future generation of pH sensors. Among these metal oxide materials, magnetron sputtered RuO_2_ thin films show outstanding properties with near Nernstian pH sensitivity, high conductivity, and excellent mechanical strength and corrosion resistance [2,3,4,5], as such it has been researched extensively.

Magnetron sputtering is one of the most attractive physical vapor deposition processes (PVD) for an extensive range of metal oxide materials due to its outstanding advantages, such as high deposition rate, excellent reproducibility, high density, and good quality of deposited thin films [5]. Many researchers have reported excellent pH response for radio frequency (RF) sputtered RuO_2_ thin films [6,7,8]. There have been many studies on the effects of different sputtering conditions on the properties and pH performance of RF sputtered RuO_2_ thin films, such as sputtering temperatures in the range of room temperature to 500 °C [9,10], deposition time/thin film thickness [11], and Ar/O ratios ranging between 10 and 2.3 [12,13]. However, there are few reports comparing the differences between direct current (DC) and RF sputtering of RuO_2_ thin films and their characteristics and pH response. Furthermore, the effects of different cathodes (either metallic or metal oxide) have not been investigated extensively.

In this study, RuO_2_ thin films were sputtered under different sputtering conditions, including DC/RF discharges, metallic/oxide cathodes (Ru/RuO_2_), different substrate temperatures (100 °C, 150 °C and 250 °C), sputtering pressures (1.0 Pa and 2.0 Pa), and thicknesses (~200 nm and ~600 nm). The structure and properties of the deposited RuO_2_ thin films studied include surface morphology, residual stress, crystal structure, composition, hardness, and elastic modulus. In addition, electrochemical experiments were conducted in terms of pH sensitivity and pH stability. The effect of different sputtering conditions on the RuO_2_ thin film properties, especially the effect of different discharges and different cathodes, were studied. Furthermore, the effect of the presence of RuO_3_ in these RuO_2_ thin films and the correlation between the pH response and—structural characteristics are discussed.

## 2. Materials and Methods

### 2.1. Thin Film Fabrication

RuO_2_ thin films were deposited using a magnetron sputtering system equipped with an axial turret magnetron head and power supply (AJA DCXS-750, Scituate, MA, USA). The turret head was mounted vertically in the bottom of the vacuum system. The target materials used for sputtering were metallic (Ru) cathode and oxide (RuO_2_) cathodes of high purity (>99.0%). The nominal size of the sputter targets was 50 mm in diameter and the distance between the cathodes and the substrate was set at 60 mm. The films were deposited onto (100) conducting silicon wafers with a resistivity of 0.05 Ω-cm. The substrates size was 25.0 mm × 25.0 mm. The substrates were electrically grounded. The deposition system was equipped with rotary and cryogenic pumps and a controlled gas introduction system. A base pressure of 1 × 10^−4^ Pa was attained in the chamber before the deposition. The oxygen reactive gas and argon inert gas were introduced into the chamber depending on the sputtering target used. The deposition pressure could be set independently of the gas flow by adjusting a throttle valve. The RF or DC powers were set at 100 W or 125 W, respectively. The deposition times varied between 15 to 40 min. The film thickness variation across the substrate was in order of 10.0%. Three groups of RuO_2_ thin films were prepared as outlined in Table 1.

Samples S1–S3 were deposited by reactive RF sputtering from a metallic cathode target (Ru) with a fixed oxygen partial pressure (Ar/O_2_ ratio of 4/1), total pressure (2.0 Pa), RF power (125 W), and deposition time (20 min). The substrate temperature was varied from 100 to 250 °C.Samples D1–D4 were sputtered from an oxide cathode target (RuO_2_). D1 and D2 were deposited by DC sputtering with fixed total pressure (2.0 Pa), DC power (100 W), and deposition time (15 min). The substrate temperatures were room temperature and 250 °C respectively. D3 and D4 were deposited by rf sputtering with fixed substrate temperature (250 °C), RF power (100 W) and deposition time (20 min). The deposition pressures were 2.0 and 1.0 Pa, respectively.Thicker samples, T1 and T2, were deposited by DC sputtering from an oxide cathode target (RuO_2_) with a fixed total pressure (2.0 Pa), DC power (100 W) and longer deposition time (40 min). The substrate temperatures were 150 and 250 °C, respectively.

### 2.2. Film Characterisation

In order to measure the surface morphology of the thin films, a Bruker SPM ICON atomic force microscope (AFM) was employed to determine the roughness and the grain size. The residual stress in the thin films was calculated from the bending height that was measured by a Dektak 3030 surface profilometer and the thickness was determined by scanning electron microscopy (SEM) (ZEISS AURIGA) from the cross-section of the films.

X-ray photoelectron spectroscopy (XPS) measurements of all samples were performed in a SPECS SAGE 150 XPS System using Mg Kα radiation at 10 kV and 10 mA. The system operated at 100 W. The base pressure in the sample chamber of the spectrometer was <1 × 10^−7^ mbar. The pass energy was 30 eV with a step size of 0.5 eV for broad scanning and was 20 eV with a step size of 0.1 eV for high resolution scanning. The instrumental resolution was 1.3 eV as measured from the FWHM of the 4f_7/2_ line for Au at 84.0 eV. Curve fitting of the high-resolution scans and peak area calculations were carried out using Casa XPS software.

Raman scattering measurements were made using a confocal RENISHAW inVia instrument in back-scattering geometry. A solid-state laser with a wavelength of 514 nm was used as the excitation source. The laser power was 1.4 mW and the laser beam was normal to the sample surface. The laser light was focused to a spot size of about 700 nm in diameter onto the sample with an optical microscope. An exposure time of 20 s was used. The signal was detected by a charge coupled device camera and a 2400 lines/mm monochromator. The resolution of the system was about 1 cm^−1^.

The crystal structure of the RuO_2_ thin films was characterised by conventional θ−2θ X-ray diffraction using an Empyrean XRD Diffractometer with a Cu Kα (λ = 1.5406 A) source. The hardness and elastic modulus of the thin films were determined by nanoindentation tests with a Hysitron Triboindenter TI 900 using a standard Berkovich indenter. The loading force was set as 2000 μN for all samples. The pH sensitivity and stability of the RuO_2_ thin films was measured using the open circuit potential (OCP) method in a commercial pH buffer solution (Merck) of different pH values (pH = 2, 4, 7, 10) with a large input resistance.

## 3. Results and Discussion

### 3.1. Surface Morphology

The AFM analysis of samples in the S group and D group are shown in Figure 1 and Figure 2. The roughness values, R_a_, for these samples are listed in Table 2. The roughness is assumed to increase as the grains tend to coarsen with increasing substrate temperature. From Figure 1b, the sample deposited at 150 °C shows deep voids and protruding grains which leads to a higher roughness than that for sample S3. However, based on Figure 1c, the thin film deposited at 250 °C is denser than that deposited at 150 °C, which is expected in accordance with the higher temperature. The likely reason behind this is that increasing the substrate temperature significantly enhances the lateral mobility of the condensing sputtered target atoms. Temperature-activated incoming depositing atoms tend to fill up the voids instead of self-shadowing, which in turn densifies the film material.

In the sample D group, when comparing the AFM images for samples D2 and D3 (Figure 2a,b), the RF sputtered thin film has a rougher surface and larger grain size than the DC sputtered thin film. This is due to the difference in the sputtering plasma. The applied potential will vary significantly between RF and DC sputter deposition. One of the major influences on the film morphology will be the bombardment of the film by high energy (charged) species. The ions within the plasma have higher energy during RF sputtering compared to DC sputtering, which is beneficial for grain growth during deposition [14]. The grain coarsening leads to a higher surface roughness. Samples D3 and D4 in Figure 2b,c indicate that the RuO_2_ thin film has a smoother surface and smaller grain size when deposited at a lower sputtering pressure. Similar behaviour was also observed in the case of DC sputtered iridium oxide films deposited onto Si substrates [15]. Samples in the T group (data not shown for brevity) exhibit a relatively low roughness as they were DC sputtered.

There is a significant difference in the roughness between S group sputtered from a metallic cathode (Figure 1c) and D group sputtered from an oxide cathode (Figure 2a). This is because thin films sputtered from the metallic cathode were in an oxygen environment. The oxidation reaction occurs near the Si substrate.

### 3.2. XPS Analysis

All peaks in the spectra were charge corrected with respect to the C 1s peak (284.6 eV). The Ru doublet peak (Ru 3d_3/2_ and Ru 3d_5/2_) and Ru 3p peaks were identified in the wide spectral scan. Since the adventitious C 1s peak at 284.6 eV coincides with the Ru 3d_3/2_ peak, the use of the Ru 3d peaks for qualitative and quantitative analysis is not reliable, but there have been a few studies using the assignment of Ru 3p peaks for ruthenium oxides, especially for RuO_3_ peaks [16]. Thus, fitting of the Ru 3d peaks is used for identification, while the Ru 3p_3/2_ peak fitting is used for ratio calculations. Figure 3 shows the peak fitted spectrum for both O 1s and Ru 3d for sample S1.

From the peak fitting analysis for both the O 1s and Ru 3d peaks of sample S1 (Figure 3), two types of ruthenium oxides are identified that contribute to the structure of the deposited thin films. The peak fittings of all samples are similar, indicating that all the samples consist of two different oxide species. According to data from the published literature presented in Table 3, the reported values of RuO_2_ and RuO_3_ match the binding energies of the Ru 3d and O 1s peaks from the present XPS measurements for all samples. In this case, the ruthenium oxides present in these thin films are identified as RuO_2_ and RuO_3_.

The peak fitting for Ru 3p_3/2_ has approximately the same FWHM for both RuO_2_ and RuO_3_ (3.4 eV). The results of RuO_2_/RuO_3_ ratios of all samples are listed in Table 4. The results show that the RuO_2_ is the dominant oxide species in the thin films deposited at the lower substrate temperatures and especially in the non-oxygen sputtering environment. An obvious trend can be observed according to the samples in the S group that were reactive RF sputtered in an O_2_ environment, that is, the percentage of RuO_3_ increases as the substrate temperature increases.

### 3.3. Residual Stress and Raman Spectroscopy

The results of the residual stresses in the thin films were calculated using the Stoney’s formula [20], which are listed in Table 5. The residual stress normally increases with increasing substrate temperature as the thin film becomes denser. In the S group, the compressive residual stress increases at first, but then decreases and converts to tensile stress with a further increase in temperature. The higher compressive residual stresses at higher substrate temperatures may be related to the denser microstructure that formed. However, greater grain growth at higher temperatures may also contribute to stress relaxation [21]. In addition, the tensile stress can linearly increase from the thermal mismatch between the thin film and the substrate with increasing deposition temperature [21]. In the case of sample S3, that was deposited at the highest temperature, grains coarsen, and the thermal tensile stress may exceed the intrinsic compressive stress, which resulted in a tensile residual stress in this thin film. The effect of sputtering pressure on the residual stress can also be observed from samples D3 and D4. At a lower sputtering pressure of 1.0 Pa, the compressive residual stress is higher in this thin film. The reason is that the lower sputtering pressure can give a lower sputtering rate, which is beneficial to deposit a denser thin film.

The Raman spectra for samples in groups S and D are shown in Figure 4**.** Three Raman-active modes, e.g., A1g and B2g can be observed in the Raman spectra and analysis of spectra for these films can provide insight into the phase constitution of each sample. The B1g mode is too weak to be observed. For the thinner films, there is a combination of RuO_2_ and Si peaks at the, e.g., mode region for each sample. The possible reason is that the RuO_2_ thin film is deposited with a nm-scale thickness onto the Si substrate, so the incident laser penetrates the RuO_2_ thin film and interacts with the Si substrate. In this case, peak fitting is applied to specify the position of the, e.g., frequency mode and is also used to assign the peak position of the A1g and B2g modes for RuO_2_.

The three major Raman-active modes, e.g., A1g and B2g for single-crystal RuO_2_ are located at 528, 644, and 716 cm^−1^, respectively [22,23]. In this experiment, all the three Raman-active modes show a red shift in the peak location. The red shift has a linear relationship with the residual stress according to Meng and Dos Santos [24]. In fact, the stress should have an influence on all modes, but the stress effect on A1g mode is greater than the others [24]. Figure 5 shows the relationship between the A1g peak position and the residual stress of the S and D groups. In the case of the group D samples, films D1, D2, and D3 show that the Raman shift increases with the residual stress linearly. Sample D4 (stress = 0.82 GPa) falls out of the trend as it is the only sample deposited at a lower pressure (1.0 Pa) than the other three samples deposited at 2.0 Pa. The deposition pressure influences the bombardment energy of the depositing atoms resulting in the modifications of the properties of the films such as texture, morphology, composition and stress. The data of the group S show a non-linear relationship. This difference in the trend can be attributed to the surface morphologies and film texture between the two groups of samples. In addition, the increasing percentage of RuO_3_ for group S sputtered from the metallic cathode that alters the O/Ru ratio. Parker et al. [25] indicated that the Raman peak positions are dependent on the O/Ti ratio in TiO_2_ films.

Chan et al. [26] assigned the peak at 800 cm^−1^ to RuO_3_ using surface-enhanced Raman spectroscopy. However, here the Raman results do not indicate a peak at 800 cm^−1^. This is likely because the region of the composition that XPS measures is only the near surface region of the thin film, while the region of the structure that Raman spectroscopy measures is much deeper into the thin film and hence the substrate (that is, a strong Si peak can be observed in the Raman spectra). The surface region may consist of RuO_2_ and RuO_3_, while the bulk region of thin film may only consist of RuO_2_. In this case, the RuO_2_ thin film can be considered as a layered structure. This is in agreement with the work of Chou et al. [8].

### 3.4. X-ray Diffraction Analysis

From Figure 6, for films deposited (S group) at both 100 and 150 °C, the preferred crystal orientation of RuO_2_ thin film is (101). With increasing substrate temperature up to 250 °C, there is a peak at (110) which increases in intensity with temperature. This result is similar to that recorded previously [27,28,29], where the RuO_2_ thin films show a preferred orientation along the (101) over the temperature range from 100 °C to 300 °C. There is a relatively weak (200) peak at the highest temperature of 250 °C, which was reported to increase with the temperature increases to above 300 °C [23]. All observed peaks of the RuO_2_ films were assigned to the tetragonal rutile structure with lattice parameters of c = 4.50 Å and c = 3.05 Å.

For the sample prepared by RF sputtering at the lower pressure of 1.0 Pa, the thin film exhibits relatively weak crystallinity (Figure 7a). From Figure 7b, the thin film prepared by DC sputtering shows a polycrystalline structure when compared to that prepared by RF sputtering. The rf sputtered thin films exhibit the (101) preferential orientation. As the temperature increases to 250 °C, there is a change in preferential orientation from (101) to (110). For the sample prepared by DC sputtering at room temperature, (not shown here) the thin film exhibits a dominant crystal orientation of (101), which is different from the reported results where the RuO_2_ samples, prepared by RF sputtering, showed an amorphous structure at room temperature [27,28]. The XRD patterned of DC sputtered film deposited at 150 °C (T2) is shown in Figure 7c, exhibiting a dominant crystal orientation of (101), A broad peak was located at around 55° that indicates a short-range periodic arrangement of RuO_2_ (211). Compared to the samples sputtered from the metallic Ru cathode, which have polycrystalline structure (Figure 6 (S3) and Figure 7b (D3)), the thin films prepared from the RuO_2_ oxide cathode have a preferential (101) orientation. This is presumably because the oxidation reaction of the metal target atoms at the substrate causes the change in the deposition behaviour of the oxygen species. Compared to the sample prepared under the same condition, but with a thinner film (Figure 7b (D2) and (Figure 7c (T2)), the thicker thin film shows a strong preferential (101) orientation. In the XRD measurements, no peaks of the RuO_3_ phase were observed for all the samples studied here. As indicated earlier, the RuO_3_ phase was identified with the XPS technique which is a surface sensitive technique (~5 to 10 nm depth) where the sensitivity depth of the XRD analysis is in the micrometers range. Therefore, due to the different depth sensitivity of these techniques, one may obtain different information if the film is not homogenous with thickness. We deduce that the RuO_3_ is present only on the outer surface of the films and not in the bulk since no XRD peaks assigned to RuO_3_ were observed.

### 3.5. Hardness and Modulus Results

The load-unload curves for samples in the D group and T group all show good homogeneity. Figure 8 shows the load-unload curves for sample D1. The load-displacement (p-h) curves for samples in the S group, however, show a greater variation, which is mainly because of the non-ideal sample surface for these samples. The high roughness and the relatively small thickness may also contribute to this effect. In this case, the deviations of the data are large, and the results are unreliable. The hardness and elastic reduced modulus values for all samples are listed in Table 6.

In brief, however, the hardness increases as the substrate temperature increases because the thin films become denser at high temperature. Residual stress also influences the hardness; compressive stress makes thin films harder, while tensile stress makes thin films softer. Sample D2, that retains the highest compressive residual stress, exhibits the highest hardness. Although the hardness result for sample S3 that is under a tensile residual stress is not reliable, based on its p-h curves, it can still be observed that this thin film exhibits a relatively low hardness. The low hardness of all samples in group S may be attributed to the greater grain size and the presence of RuO_3_.

Sample D3, prepared by RF sputtering, is softer than sample D2 prepared by DC sputtering. This is because of the different plasma effect of RF discharges that makes the grains coarsen. Sample D4 deposited at a lower pressure exhibits a higher hardness than sample D3 due to its denser microstructure at a lower sputtering rate. The thin film hardness is highly dependent on the sputtering conditions. Búc et al. [9] reported that a sputtered RuO_2_ thin film exhibited a hardness of 9.4 ± 1.7 GPa, while Zhu et al. [29] reported that the thin film exhibited a hardness of 20.4 ± 2.4 GPa. The hardness results obtained in this experiment are in the range of these reported values.

### 3.6. Electrochemical Results

Figure 9 shows the potential of saturated calomel electrode (SCE) versus pH values for the different groups of samples. The pH sensitivity of all samples is listed in Table 7. Samples S1, S3, T1, and T2 show near-Nernstian slopes of 53.6, 56.4, 57.4, and 54.1 mV/pH, respectively.

The poor pH sensitivity for sample S2 could be attributed to its high porosity. From its AFM image, shown in Figure 1b, the surface exhibits deep voids and protruding grains indicating significant pore formation in the RuO_2_ thin film. The enhanced pore formation increases the scattering of the charge carriers and thus reduces the carrier mobility [30,31], which in turn results in the low pH sensitivity of this thin film. Furthermore, sample S2 also exhibits a potential drift. This is attributed to the trapping of hydrogen and hydroxide ions when they diffuse into the pores of the thin film [32].

When comparing the results of the pH sensitivity and the percentage of RuO_3_ in the thin film, from samples S1 to S3 and T1 to T2, it can be also seen that RuO_2_ thin films with a higher percentage of RuO_3_ have a better pH sensitivity. This is because a higher percentage of RuO_3_ leads to the higher oxygen ratio in the thin film, which increases the redox reaction speed.

In the D group, sputtered from the RuO_2_ cathode, it can be seen that thin films prepared by DC sputtering have a higher pH sensitivity than thin films prepared by RF sputtering. Both samples D3 and D4 exhibit the lowest pH sensitivity probably due to their smaller thickness, which is in agreement with the published literature [33], where the pH sensitivity of RuO_2_ electrodes decreases as their thickness decreases. This can also be seen in samples D2 and T1, which were prepared under the same sample conditions, but with different thicknesses. The pH sensitivity of the thicker sample T1 is highest. The poor pH response of the thinner RuO_2_ films indicates that there is insufficient RuO_2_ in the coating to react with the solution. This would result in the extensive exposure of the underlying Si substrate, which could form a pH dependent galvanic couple between the substrate material and solution and thus reduce the pH sensing performance.

All the samples show a stable output potential for all pH values over time, except sample S2 that exhibited a potential drift for both acidic and alkaline solutions. Figure 10 shows the potential versus time at different pH buffer solution of sample S3 as a representative example where the stability of the output is clearly evident.

## 4. Conclusions

In this study, the effects of magnetron sputtering conditions, including DC/RF discharge, metallic/oxide cathode target, different substrate temperatures, pressures, and thicknesses, on the characterisation and pH performance of the RuO_2_ thin film have been investigated.

The effects of metallic/oxide cathode. The thin films sputtered from the metallic cathode were much rougher than those sputtered from the oxide cathode due to the oxidation reaction near the substrate/as-deposited thin film and the bombardment of negative oxygen ions during sputtering process. This leads to a higher proportion of RuO_3_ in the thin film. The thin films sputtered from the oxide cathode were found to be much harder.The effects of DC/rf discharge. The RuO_2_ thin films deposited by RF sputtering were rougher than that deposited by DC sputtering. The rf discharge is beneficial to the grain growth in the thin film, which leads to a softer film. The DC sputtered thin films have a higher pH sensitivity response than the RF sputtered thin film.The effects of substrate temperature. The RuO_2_ thin film is rougher and denser at higher substrate temperatures due to greater grain growth. The compressive residual stress increases with increasing temperature. The percentage of RuO_3_ in the RuO_2_ thin film increases as the substrate temperature increases.The effects of sputtering pressure. The RuO_2_ thin film are rougher and less dense when deposited at a higher sputtering pressure due to the higher sputtering rate. At a lower pressure, the thin film retains a higher compressive residual stress, which results in a higher hardness. The percentage of RuO_3_ is higher at lower sputtering pressure.The effects of the presence of RuO_3_. The presence of RuO_3_ in the RuO_2_ thin film results in the thin film to be rougher and softer. The higher percentage of RuO_3_ in the thin film leads to a better pH response. The Raman red shift is related to both the residual stress and the O/Ru ratio.Sample S3 RF, sputtered from a metallic cathode at 250 °C, and thick sample T1, DC sputtered from the oxide cathode at 250 °C, have near-Nernstian pH sensitivities of 56.4 and 57.4 mV/pH, respectively. The RuO_2_ thin film RF sputtered from the metallic cathode at higher temperature exhibits a good pH performance with a thinner thickness. However, the thin film is rougher, less dense, and softer. The RuO_2_ thin film DC sputtered from the oxide cathode at higher temperature exhibited a good pH performance with a smoother surface, denser microstructure and higher hardness.

## Figures and Tables

**Figure 1 sensors-20-06432-f001:**
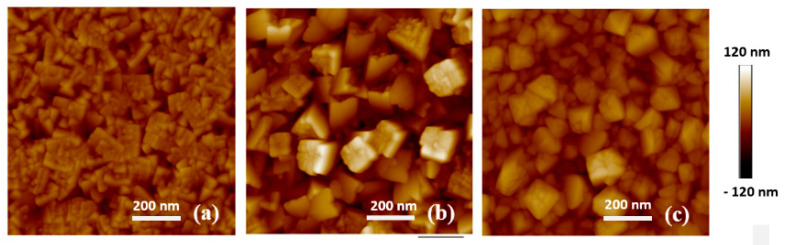
Atomic force microscope (AFM) images of the S group samples; (**a**) S1—100 °C; (**b**) S2—150 °C; (**c**) S3—250 °C.

**Figure 2 sensors-20-06432-f002:**
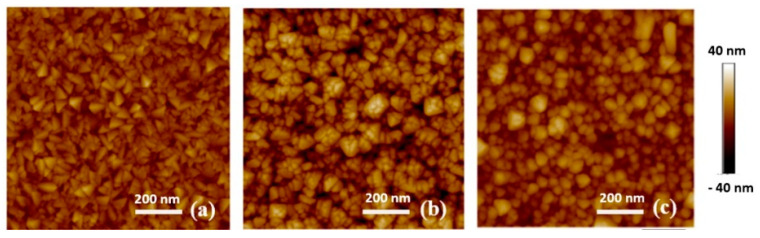
AFM images of D group samples; (**a**) D2, (**b**) D3—and (**c**) D4.

**Figure 3 sensors-20-06432-f003:**
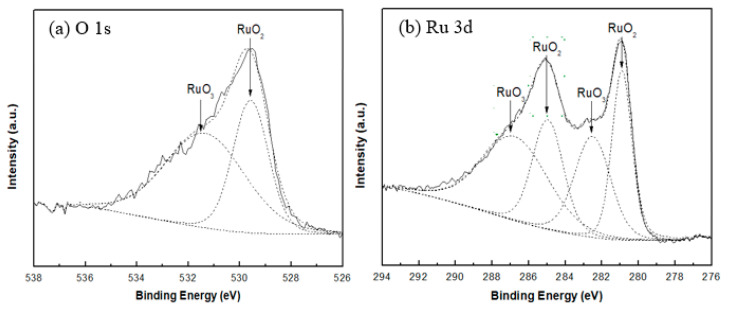
(**a**) O 1s and (**b**) Ru 3d peaks region for sample S1.

**Figure 4 sensors-20-06432-f004:**
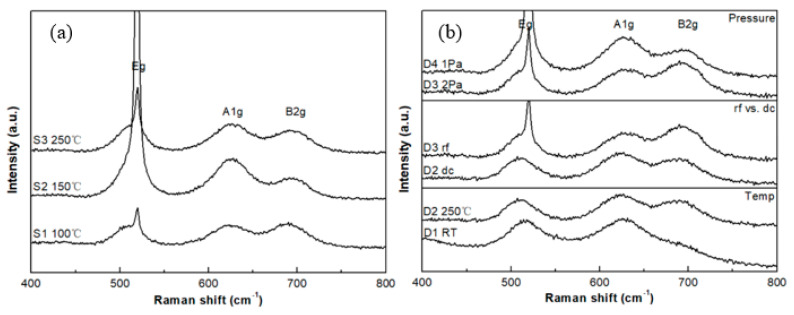
Raman spectra for samples in the (**a**) S group (**b**) D group with deposition pressure, RF and DC sputtering and deposition temperature variations.

**Figure 5 sensors-20-06432-f005:**
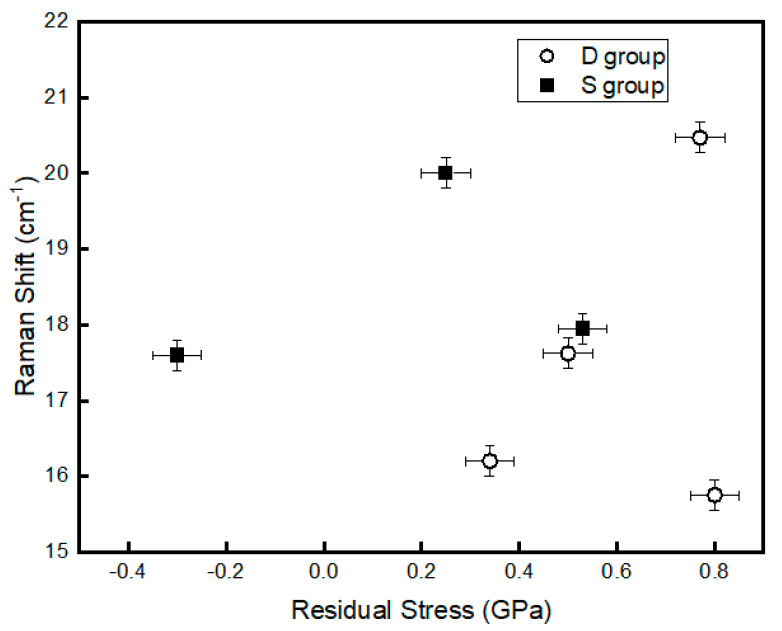
The relationship between the residual stress and Raman shift for the A1g mode.

**Figure 6 sensors-20-06432-f006:**
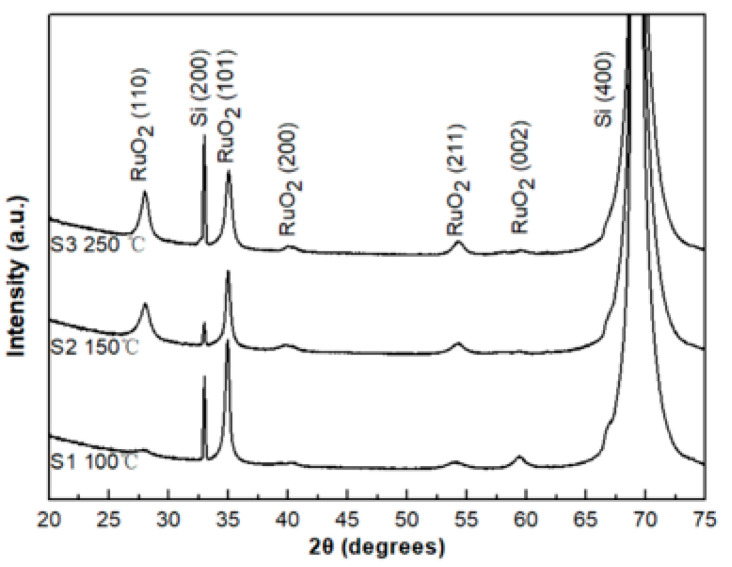
X-ray diffraction patterns of samples from the S group.

**Figure 7 sensors-20-06432-f007:**
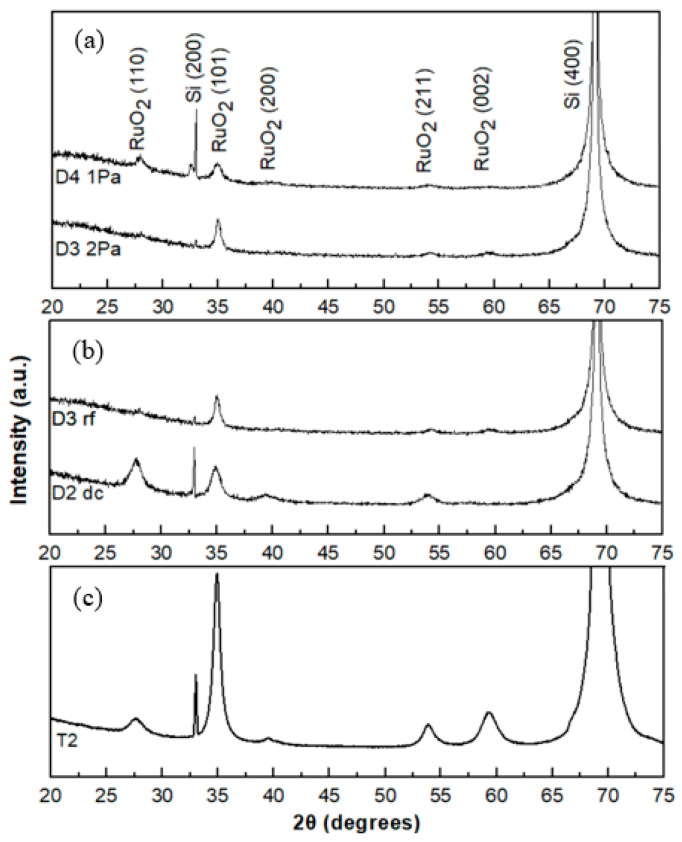
X-ray diffraction patterns of the D group samples (**a**) RF discharge: 1.0 Pa and 2.0 Pa; (**b**) DC/RF difference; and (**c**) T2 with thicker thickness (~700 nm).

**Figure 8 sensors-20-06432-f008:**
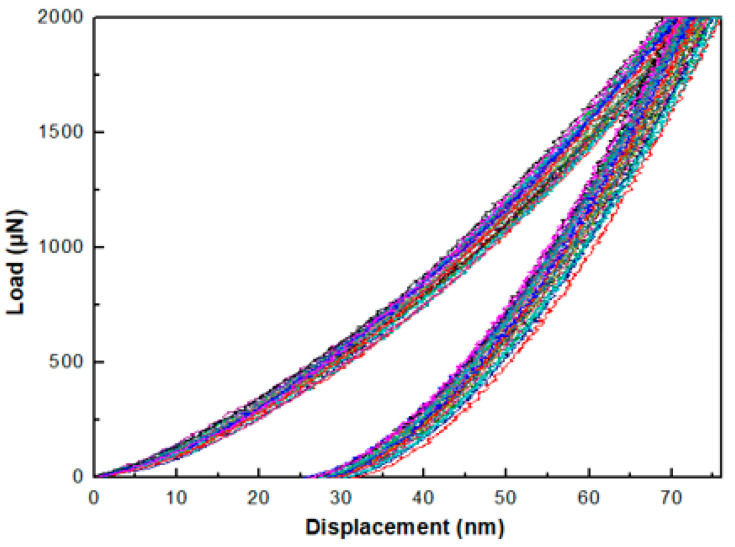
Load-unload curves of sample D1 under 2000 μN maximum loading force.

**Figure 9 sensors-20-06432-f009:**
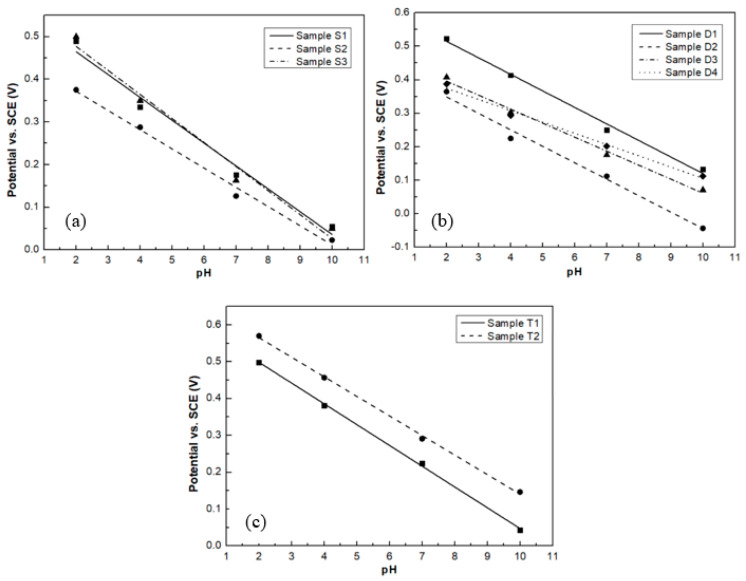
pH sensitivity of samples (**a**) S group; (**b**) D group; (**c**) T group.

**Figure 10 sensors-20-06432-f010:**
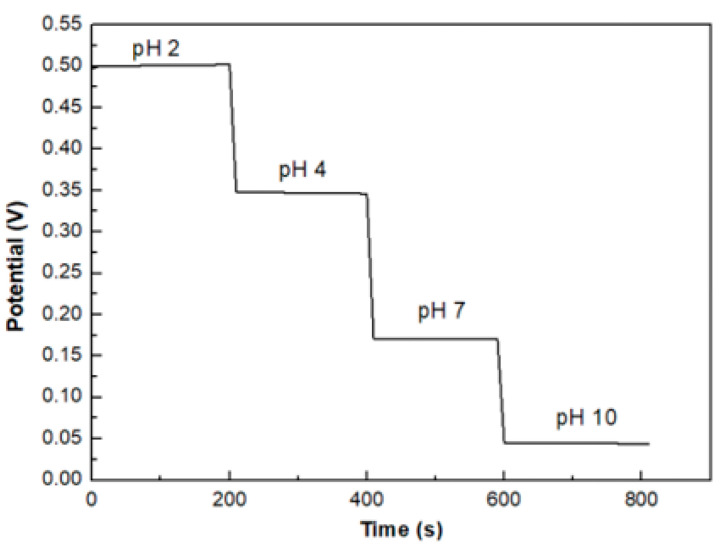
pH stability of sample S3.

**Table 1 sensors-20-06432-t001:** Deposition conditions for all samples, direct current (DC) and radio frequency (RF)

Sample	S1	S2	S3	D1	D2	D3	D4	T1	T2
DC/RF	RF	RF	RF	DC	DC	RF	RF	DC	DC
Pressure (Pa)	2	2	2	2	2	2	1	2	2
Power (W)	125	125	125	100	100	100	100	100	100
Deposition time (min)	20	20	20	15	15	20	20	40	40
Temperature (°C)	100	150	250	R.T.	250	250	250	250	150
Cathode target	Ru	Ru	Ru	RuO_2_	RuO_2_	RuO_2_	RuO_2_	RuO_2_	RuO_2_

**Table 2 sensors-20-06432-t002:** Surface roughness of the RuO_2_ thin films.

No.	S1	S2	S3	D1	D2	D3	D4	T1	T2
R_a_ (nm)	12.9	27.1	15.4	3.44	4.61	7.72	5.68	5.20	2.77

**Table 3 sensors-20-06432-t003:** XPS binding energies of different RuO_x_ compounds from the literature.

RuO_x_	O 1s (eV)	Ru 3d_5/2_ (eV)	Ru 3d_3/2_ (eV)	Ru 3p_3/2_ (eV)	Ref.
RuO_2_	528.9–529.4	280.1–281.3	284.8–285.0	462.2	[16,17,18,19,20]
RuO_3_	530.7–531.2	281.7–282.5	286.6–287.0		[16,17,18,19,20]
RuO_4_		282.6–283.3			[16,17,19]

**Table 4 sensors-20-06432-t004:** RuO_2_/RuO_3_ ratios for all samples.

No.	S1	S2	S3	D1	D2	D3	D4	T1	T2
RuO_2_/RuO_3_ ratios	1.74	2.61	3.63	4.58	3.78	3.80	3.02	2.46	3.09

**Table 5 sensors-20-06432-t005:** Residual stress for all samples.

No.	S1	S2	S3	D1	D2	D3	D4	T1	T2
σ (GPa)	0.25	0.53	−0.30	0.50	0.77	0.34	0.82	0.40	0.41

**Table 6 sensors-20-06432-t006:** Hardness and elastic reduced modulus for all samples.

Sample	Hardness (GPa)	Elastic Reduced Modulus (GPa)
S1	6.1	144.8
S2	3.8	120.9
S3	6.2	133.9
D1	13.6	164.9
D2	17.2	190.4
D3	10.3	157.8
D4	11.5	156.7
T1	12.3	176.1
T2	12.0	172.7

**Table 7 sensors-20-06432-t007:** pH sensitivity and linearity of all samples.

Sample	Sensitivity (mV/pH)	Linearity
S1	53.6	0.9735
S2	45.0	0.9882
S3	56.4	0.9708
D1	49.3	0.9907
D2	49.1	0.9835
D3	41.8	0.9896
D4	33.6	0.9846
T1	57.4	0.9986
T2	54.1	0.9980

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
