# Peer review of "Advanced RuO2 Thin Films for pH Sensing Application"

_sensors, 2020, doi:10.3390/s20226432_

Round 1
Reviewer 1 Report
The manuscript by Xinyue Y. Yao et al. shows a large amount of thin film characterization data of sputtered RuO2 films with the aim to showcase their pH sensing ability. I do not always agree with the authors conclusions and therefore can not support the publication of the work in the current form. Please make improvements as indicated below.
I am missing a comparison of the used deposition parameters to parameters used in the references. For example for sample series S1-S3: how does the Ar/O2 ratio compare to the one used by other authors? In other words I am asking the authors to consider giving a few details of the other studies in their introduction to make the reading of the present manuscript easier.
The authors write on lines 207 to 208 that the S group shows a non-linear relationship in the Raman shift shown in Figure 5. They highlight that this is in contrast with the D-group. Here the authors selectively ignore sample D4. The authors must specify reasons why sample D4 is expected to fall out of the trend of the other D group samples or change their argument in lines 209-211!
In Fig. 6 the authors show X-ray diffraction of the S group and label all peaks as originating from RuO2 only. Especially sample S3 has a lot of the RuO3 phase. Can any of the shown peaks be related to this phase or why does the high amount of RuO3 phase not show up in these measurements? It would be also advantageous if the authors would specify the crystal structure and lattice parameters of these phases so that readers can check their peak assignment.
I can not follow the arguments of the authors on lines 240-248: There is written the claim that samples sputtered from an RuO2 target have a preferential (101) orientation which is supposed to be in contrast to samples from the S group which on line 224 are said to have a preferential (101) orientation? It might be that the authors speak only about sample D3 which seems to show exclusively the (101) peak, but that's not what is written! Please fix your arguments!
Also your argument in lines 246-248 makes no sense: There must be a preferential growth of the grains with (101) orientation compared to other grains to lead to a change of the peak ratio upon a thickness change which is again not what you write: You write grains grow along [101] direction. If all grains do that, the peak intensity ratio (integrated intensities) would, however, not change.
The authors argue that the best pH sensing values are a result of presence of RuO3 and higher thickness. Can the authors explain why sample S1 with rather low thickness and the absolutely lowest amount of RuO3 shows one of the best pH sensing values?
I am also missing some essential experimental information:
*) What wafer size was used as a substrate? Since you mention the bending as a measure of the residual stress I am wondering how homogeneous this bending (and the film thickness) are across the film area?
*) The authors decided to use the scattering angle on the x-axis of figure 6 and 7. This quantity is meaningless unless the used X-ray wavelength is given.
line 182: I guess you mean the deposition temperature, please clarify this in the text.
line 210: fix the English please.
caption of Table 7: Please explain the '*' symbol you included in some lines.
Figure 9: abbreviation SCE is not introduced
Reviewer 2 Report
The manuscript is discussing the deposition and properties of magnetron sputtered thin films of RuO2. The main question of the study is the difference between the use of metallic and oxide targets as well as the use of DC and RF discharge voltages.
The manuscript describes a broad set of deposition conditions and an even broader set of characterization techniques. However, in my opinion, the manuscript is not well constructed and each characterization section is isolated from the others without any link between them. For example, the XRD patters are not correlated with the AFM morphology or the values of the residual stress. I believe that the results presented are solid with some interesting sections but the discussion and the explanation is rather poor and confusing. I would suggest a complete rewriting of the manuscript, apart from introduction and experimental sections.
Below, are shown only some examples of some rather major issues in the manuscript .
- The paragraph starting at line 131 is very difficult to understand. Text starts describing the different energy of the atoms in the RF and DC plasma, which I suppose is the plasma potential, and then they link it with higher grain growth, displaying a reference with ZnO thin films. I believe that this part and in general the understanding of the deposition dynamics is lacking and needs more work in order to link the various deposition conditions to the observed characteristics of the thin films.
- Line 143.”The as-deposited thin films…and the grain becomes larger.”. This is a very wrong statement or I have misunderstood it completely.
- In the manuscript, it is discussed that XPS revealed the presence of RuO3 and RuO2 in the films and the authors reflected this composition/ratio to the whole film. As far as I know, XPS is a very surface-sensitive technique and it does not reflect composition throughout the bulk of the film. Also, the table 3 and 4 should be omitted since no useful information (for the scope of this manuscript) is presented there.
- The discussion for the x-ray diffraction analysis is very confusing to the reader, presumably because the expressions used are unusual, such as the “multi-oriented crystal structure”, etc.
Minor issues and typos
- Usually, abbreviations are in capital letters, such as DC and RF.
- Please add details about the bias of the substrates (grounded, floating etc…)
- At line 141 figure number is incorrect.
- Figure 2 caption needs rewriting.
- Table 3 should be omitted.
- Line 176: New sentence
- Table 7 some values have a star.
Reviewer 3 Report
I thank the authors for the well organized article. They have presented nicely detailed material characterization and nicely summarized electrochemical results. Conclusions are presented in a nicely formulated way.
Here are more specific comments from me:
- In line 53, thickness should be plural as 'thicknesses', since other parameters are plural
- In line 56, 'the' before 'effects' should be erased. The is not used for plural words in general. Either make effect singular.
- In line 57, 'the' before effects. Either make effects singular.
- In line 58 same as comments 2 & 3.
- In line 59, correction can be made in two ways. Either 'between the pH response and structural characteristics' or 'between pH responses and their structural characteristics'.
- In line 167, 'is' to be changed with 'has'.
- In lines 323-324, the sentence can be rephrased due to high sensitivity at S3.
Reviewer 4 Report
The paper entitled "Advanced RuO2 Thin Films for pH Sensing Application" submitted by Y.Yao et al., presents the investigation results on RuO2 thin films deposited by utilization of magnetron sputtering technique under different sputtering conditions such as DC/RF discharges, metallic/oxides modes, different substrate temperatures, sputtering pressures, thicknesses. The experiments were well-designed and finally well-performed. The results are presented in a clear form. Minor remarks: all figures - better quality is needed; figure 5 please provide the statistics and error bars if applicable. Finally, the paper is well-organized and well-structured, the conclusions are supported by data and presented in a very systematic form, which helps readers to follow the novelty of the paper. I suggest accepting after minor revision, where Figures will be changed for better ones; the sputtering system photo or drawings will be provided as well as detailed info about the vacuum components installed in the system.
Round 2
Reviewer 1 Report
While I find the authors made some improvements and fixed obvious errors, the quality of the reply as such is rather poor.
It appears especially awkward that some questions where said to be "answered" by removing previous statements from the manuscript. The questionable statements are, however, still present in the manuscript! For other questions the changes listed in the reply letter and the text in the manuscript do not agree. More changes are required before this work can be accepted. Specific comments follow.
*) The discussion of the Raman shift on page 7 is unreadable. The sentence which was changed does not make sense and is much too long. Please rewrite this discussion!
*) The authors argue in the reply letter: """The RuO 3 phase was observed with the XPS analysis only, we believe this phase was observed on the outer surface of the films, hence the X-ray diffraction data has not detected the RuO 3 phase.""" Its true that XPS and XRD have different depth sensitivities, but if both phases are crystalline than XRD should detect them! So the real difference can only be in the crystallinity/grain size of these phases. A proper discussion of the results which considers data of the various shown methods should include explanations why only certain parts of the films are detected by the methods!
*) I have previously criticized the discussion of the XRD data. Unfortunately the revised text still does not make any sense. When I look at the data shown in Fig. 6 and 7 I do not see what the authors want to say with the following sentences on page 8/9:
"""From Fig. 7 (b), the thin film prepared by DC sputtering shows a polycrystalline structure when compared to that prepared by RF sputtering.""" All your films show polycrystalline diffraction patterns with various degrees of crystallinity and different preferred orientation.
Also """For the sample prepared by DC sputtering at room temperature, shown in Fig. 7 (c), ...""" Fig. 7c does not show data from sample D1 and therefore was not grown at room temp. Diffraction patterns of D1 are not shown at all!
Also: """As the temperature increases to 250 °C, there is a change in preferential orientation from (110) to (101)""" From what you see this? Samples from the S series where grown at different temp and show quite the opposite trend to what you describe!
Remove the use of "multi-oriented crystal structure" as requested also by another reviewer.
*) The authors claimed in their reply that the statement on page 9: """This is because grains are growing along [101] direction under a longer deposition time, and the thicker the thin film the more atomic layers can diffract X-rays""" was removed. I do, however, find it unchanged in the revised version of the manuscript!
Reviewer 2 Report
The new version of the manuscript has been lightly revised only in some specific sections but without significant changes. My previous arguments still apply; each section is presented and discussed in an unclear way and almost independently from each other. According to the size of the manuscript and to the data shown, authors have done a lot of experimental work, gathering solid data but the discussion is rather poor, and sometimes confusing.
In this form, the manuscript is lacking scientific soundness and needs to be revised. In my opinion, in order the manuscript to be considered for publication, I would suggest an extensive revision of section 3.
